# Psychological Intervention in Women Victims of Childhood Sexual Abuse: An Open Study—Protocol of a Randomized Controlled Clinical Trial Comparing EMDR Psychotherapy and Trauma-Based Cognitive Therapy

**DOI:** 10.3390/ijerph19127468

**Published:** 2022-06-17

**Authors:** Milagros Molero-Zafra, María Teresa Mitjans-Lafont, María Jesús Hernández-Jiménez, Marián Pérez-Marín

**Affiliations:** 1Faculty of Health Sciences, Valencian International University-VIU, 46002 Valencia, Spain; milagros.molero@campusviu.es (M.M.-Z.); mariateresa.mitjans@campusviu.es (M.T.M.-L.); mariajesus.hernandez@campusviu.es (M.J.H.-J.); 2Department of Personality, Assessment and Psychological Treatments, Faculty of Psychology, University of Valencia, 46010 Valencia, Spain

**Keywords:** childhood sexual abuse, women, EMDR, TF-CBT, randomized clinical trial

## Abstract

Introduction: Most victims of sexual abuse have symptoms that may lead to post-traumatic stress disorder. This study aims to offer evidence-based psychological treatment to women who have been sexually abused earlier in life and currently have sequelae from that trauma. With this treatment, each of the women in the study will hopefully improve their overall quality of life and, more specifically, it is expected that post-traumatic stress symptoms will decrease, as found in recent studies, as well as strengthening their security, confidence, and coping with the situations they have experienced. Methods and analysis: The effect of two therapeutic approaches focused on the improvement of trauma will be evaluated in a sample of 30–50 women victims of childhood sexual abuse, with a randomized clinical trial comparing EMDR psychotherapy and trauma-focused cognitive behavioral therapy. According to the literature reviewed, both approaches will considerably improve self-esteem when the appropriate number of sessions are conducted, significantly reducing general psychiatric symptoms and depression. Furthermore, the effects are sustained over time. It should be noted that this study will be carried out comparing both therapies, analyzing both the differential benefit of each and the cumulative effect of receiving both treatments and in which order. It is also intended to demonstrate that implementing the protocols presented in this study will help improve the quality of life of the women who benefit from them, and after this study, it will be possible to replicate this program in other people with the same problems. Each of the therapeutic benefits of each of them will be analyzed, and clinical and logistical guidance will be provided to implement both, including a session-by-session protocol.

## 1. Introduction

Some female victims of childhood sexual abuse do display resilience; however, the majority often suffer devastating consequences, resulting in a highly unsatisfactory life experience [1]. Female victims of childhood sexual abuse often experience significant clinical symptoms related to anger and rumination, along with low levels of emotional regulation strategies and low levels of self-esteem [2,3,4,5]. The presence of clinical psychopathology is also a common element in this group. Studies report that childhood sexual abuse is associated with the presence of generalized anxiety, attention deficit and hyperactivity symptoms, major depressive symptoms, antisocial personality, and substance abuse problems [6,7].

The long-term psychological effects of sexual abuse are understood in a developmental context in terms of post-traumatic stress disorder (PTSD). Research on female victims of childhood sexual abuse shows increased dissociation, sleep disturbances, tension, sexual problems, and anger, as well as increased use of psychoactive substances and more frequent reports of suicide attempts, substance addiction, and re-victimization [8]. Emotional regulation skills are often impaired among these women and are risk factors for both internalized and externalized behavioral problems, dysregulation, and dissociation [9].

The effects of childhood sexual abuse affect physical, psychological, and social functioning. They generate significant psychopathological alterations such as emotional alterations, post-traumatic stress disorder, depression, sexual dysfunctions, perceived sexual dissatisfaction, and difficulties in self-acceptance of one’s body [10,11,12,13,14]. Specific interventions have proven to be effective in helping people cope with traumatic experiences.

On the one hand, trauma-focused cognitive behavioral therapy (TF-CBT) is an evidence-based treatment model designed to help patients (children, adolescents, and young people) and their families to overcome the symptoms resulting from exposure to a traumatic experience [15]. After numerous investigations that support it, it is currently considered a well-established intervention to treat post-traumatic stress and associated symptoms [16,17,18].

TF-CBT is an evidence-based model for treating trauma in childhood and adolescence that comprehensively includes parents or non-offending caregivers throughout treatment [19]. Its efficacy has been proven in several randomized controlled trials [20,21]; these researchers evaluate the effectiveness of TF-CBT in community settings compared to other therapeutic approaches and report that individuals receiving TF-CBT report significantly lower levels of post-traumatic stress symptoms, depression, and general mental health symptoms compared to those from other groups. Children and adolescents who received TF-CBT showed a significant improvement in functional impairment. Other findings indicate that the therapy effectively treated trauma-related symptoms and improved psychosocial functioning in children and adolescents [22,23]. A study evaluating both chronicity of trauma and the development of PTSD in the context of evidence-based treatment supports the efficacy of TF-CBT in treating PTSD in women who have suffered acute and chronic trauma [24].

On the other hand, Shapiro’s eye movement desensitization and reprocessing (EMDR) psychotherapy [25,26] is one of the transdiagnostic psychotherapeutic models of choice in international guidelines addressing the sequelae of traumatic life experiences. Different international clinical associations recognize this trauma-focused psychotherapeutic approach for dealing with trauma, adverse life experiences, and psychological stressors. The World Health Organization (WHO) includes it among the recommended treatments for post-traumatic stress [27]. This therapeutic modality incorporates bilateral attention stimulation (BLS)—through eye movement saccades or tactile or auditory stimulation (tapping). A specific component of this type of psychotherapeutic approach is combined with past events related to present discomfort and the triggers of the present, and the preparation of the patient to face future situations.

Since Shapiro published her first randomized study in the late 1980s [25], where EMDR was shown to be effective in reducing PTSD-related symptoms, studies on the efficacy of this therapeutic model have increased, both for this pathology and other clinical conditions [28,29,30], with an interesting study on the long-term effect of EMDR in female victims of childhood sexual abuse [31,32] and, more recently [33], a study showing the efficacy of EMDR therapy in the treatment of post-traumatic stress disorder derived from childhood sexual abuse. In a recently published systematic review [34], the results of 87 randomized controlled clinical trials were organized into three broad categories according to different explanatory hypotheses about the efficacy of EMDR. Regarding the role of eye movement, it has been concluded that eye movement adds an effect to the treatment. The effect size is large and significant when analyzing only the movements themselves and moderate and significant when included within a clinical treatment [35]. One of the main advantages of EMDR is that both its standard and specific protocols for treating different psychological or psychiatric problems are well standardized and scientifically validated, which facilitates both its clinical use and scientific research [36]. Over the last decade, there has been a large increase in scientific publications comparing the efficacy of EMDR with other well-established and accepted psychological therapies, including cognitive behavioral therapy (CBT), exposure therapy, and narrative therapy.

Overall, trauma-focused therapies are shown to be effective in treating PTSD in adults who have been sexually abused during childhood [37], which is why both approaches described above and with scientifically supported clinical evidence have been included in the present study.

Thus, both therapeutic approaches, EMDR and TF-CBT, can be useful to intervene effectively in adult subjects who present significant symptoms after suffering childhood/adolescence trauma from being sexually abused [38]. In the meta-analysis by Khan et al. [39], whose objective was to compare the efficacy of CBT and EMDR in (i) alleviating post-traumatic symptoms and (ii) alleviating anxiety and depression in patients with PTSD, after analyzing 11 studies with a total sample of 547 participants, results displayed a greater effect (more significant improvements) of EMDR on CBT, both reducing post-traumatic symptoms (SDM (95% CI) = −0.43 (−0.73–−0.12), *p* = 0.006) and anxiety (SDM (95% CI) = −0.71 (−1.21–−0.21), *p* = 0.005). However, no significant differences were found between CBT and EMDR in the reduction of depression (SDM (95% CI) = −0.21 (−0.44–0.02), *p* = 0.08). Hence, both therapeutic approaches can be useful to intervene effectively in adult subjects that present significant symptoms after trauma suffered in childhood/adolescence from being sexually abused [38].

Moreover, it has been observed that during the COVID-19 pandemic, the psychological treatments delivered online have increased exponentially due to the benefits they offer in terms of reducing infection risks [40]. This is of great relevance since the impact of the COVID-19 pandemic and its restrictions have increased the prevalence of mental disorders and worsened those mental disorders that existed before the pandemic [41]. Fortunately, the emergence of new technologies has provided a variety of tools that can be used, especially during the current COVID-19 pandemic situation, such as the MHealth, which is defined as “*the practice of medicine and public health supported by mobile devices such as telephones, patient monitoring devices, digital assistants and other wireless devices*” [42]. MHealth can be an opportunity to improve social relationships in this group and increase their social support while working on improving physical and emotional health. Evidence is available about the effectiveness of the TF-CBT [43] and the EMDR [44,45] therapies delivered in an online format.

This study aims to address the impact on adult victims of trauma related to sexual abuse in childhood and adolescence, proposing the design and implementation of two group psychological treatment programs based on EMDR and TF-CBT, both conducted online. This study will be carried out by comparing both therapies, analyzing both the differential benefit of each one and the cumulative effect of receiving both treatments and in which order. In the design of this study’s intervention protocol and with the proposed trauma-focused treatments, the aim is to promote emotional regulation in order to reduce dissociation and favor the adaptation of abused women.

## 2. Method and Analysis

Following the CONSORT statement [46], a randomized clinical trial (RCT) is proposed, with no control group, and the allocation to two types of treatment, (1) TF-CBT and (2) EMDR. After the initial evaluation (T1), the participants will be randomly assigned to one of the two treatments.

A well-designed and properly conducted randomized controlled trial (RCT) is essential for assessing the effectiveness of new treatments. However, besides good scientific and methodological expertise, adequate RCT reporting is necessary to allow readers to assess the validity of the trial’s results. The reporting of RCTs is facilitated by guidelines such as the Consolidated Statement on Reporting Trials (CONSORT). The CONSORT checklist for trial reporting provides a template for authors, reviewers, and editors to ensure consistency and improve the quality of trial reporting. One of the elements of the CONSORT checklist is the flow chart, which describes the flow of participants through the different trial phases, i.e., eligibility assessment, recruitment, intervention allocation, follow-up, and data analysis. This information is key to determining the validity of the trial results. The number of eligible trial participants provides information on the generalizability of the results (external validity). The number of participants available at each stage of the trial ensures that the exclusion of participants at any stage is not biased (internal validity).

The schedule of enrolment, interventions, and assessments (SPIRIT schedule) can be found in Table 1. This schematic diagram aims to represent the overall schedule and time allocation of the trial participants in an effective way.

The randomization of the participants will be carried out with the Randomizer.org program. The randomization process can be found in Figure 1.

ADR generated the randomization, MJHJ registered the participants, and MJHJ allocated the participants to the two groups.

In the case of kinship relations between participants in the same group, a further condition for non-overlapping will be carried out with these participants.

The participants will be unaware of the existence of the two experimental conditions, and therefore will not be notified of the application procedure and specific assignment to each of the treatment groups. However, the institutions involved in the recruitment of participants and the therapists and members of the research ethics committee that endorse the study will be aware of the existence of the two intervention groups.

The study is aimed at adult victims of sexual abuse during childhood. All women in the participant sample will benefit from both treatments. We will not use an untreated control group or waiting list because, in adult victims of childhood sexual abuse, the symptoms tend to become chronic, and the probability of spontaneous recovery is low [47,48]. Considering that it is not possible to leave participants without treatment and that this project will be carried out in associations such as ACASI (Association Against Child Sexual Abuse), no statistical power calculation is made, as treatment will be offered to all participants since not receiving specialized care could negatively affect patients.

Subsequently, after allowing the same amount of time for each type of intervention (TF-CBT or EMDR) (8 weeks), participants will be reassessed to measure changes without receiving the treatment (T2). After this second assessment, they will receive the treatment to which they were initially assigned after randomization. After receiving the first treatment, participants will be reassessed (T3). After this assessment, participants will receive the other treatment. Finally, after completing the second intervention, they will be assessed again (T4).

In this way, it will be possible to evaluate both the effectiveness of each of the protocols separately (intragroup comparison of each treatment condition), as well as the level of effectiveness, obtained by comparing both treatment groups (intergroup comparison, after the application of each type of intervention). The design will also provide evidence of the efficacy and benefits of the consecutive application of two protocols, which would make both types of psychotherapy equal in efficiency and effectiveness. In addition, it will provide results on the possible beneficial contribution of receiving a more extended treatment (the two interventions consecutively) versus only one of them (in case the T4 data significantly improve the health and well-being of the subjects compared to the T3 data). The evaluation phases will be at 0 months (T1), after eight weeks (T2), after eight weeks (T3), and after eight weeks (T4). The study design is detailed further in Figure 2.

Analyses will be conducted to ensure that both groups are homogeneously comparable. Our starting hypotheses state that, after receiving one or the sequence of the two intervention protocols, a statistically significant increase (*p* < 0.05) in life satisfaction and self-esteem and a statistically significant reduction (*p* < 0.05) in the psychopathological symptoms will be observed. We also hope to find more significant therapeutic benefits in subjects who first received EMDR psychotherapy (both when initially comparing EMDR versus TF-CBT and after comparing EMDR + TF-CBT versus TF-CBT + EMDR).

## 3. Procedure, Recruitment Process, and Participants

This study will be carried out in collaboration with associations from the area of study that will facilitate the recruitment of participants. In particular, the sample will be accessed by collaborating with the ACASI association. For the initial study sample, the starting point was one of the reference organizations on this issue in this region: ACASI, the Asociación Contra el Abuso Sexual Infantil (Association Against Child Sexual Abuse). This organization has extensive experience in supporting women with this specific problem in Valencia and has been operating since the beginning of 2005. Its aim is to inform and raise awareness about the consequences or sequelae of childhood sexual abuse. It also offers legal and psychological counselling, as well as guidance to family members and friends of child sexual abuse victims. Therefore, given the high correlation between the objectives of the organization and those of this study, as well as the degree of alignment between the profiles of the users of this organization and the inclusion criteria for participation, it was decided to initiate the project with a sample of participants from this association. The representatives of this organization who are in charge of the reception of women victims of sexual abuse will proceed to inform the women about the existence of this study and offer them participation in it. After this, those women who have expressed their wish to participate in the study to those responsible for ACASI will be contacted by the members of the research team to establish contacts and formal agreements to be able to form part of the study sample (patient information sheet, confidentiality agreements, etc.).

The therapy sessions in both intervention protocols will be conducted in an online group format through the Zoom videoconferencing platform. The sessions will have an estimated duration of 60 min. The frequency will be once per week. Each of the therapy protocols consists of eight treatment sessions. The groups will be composed of a minimum of four women and a maximum of six to have a greater containment capacity given the telematic format.

If any of the participants have not been able to regulate their emotional state after the work with the traumatic event in the group session, both in one or the other intervention, individual attention will be provided, and a virtual room will be created for this purpose. This will be a stabilization intervention.

To be included in the final study sample, participants must have completed a minimum of 75% of each intervention protocol, and have participated voluntarily without any incentive.

The intervention program will first be applied to a pilot sample of 4–6 patients from each treatment group. Subsequently, modifications will be made to the program based on the results obtained and the program will be applied to women who meet the inclusion criteria and regularly attended non-profit associations where psychological care is provided to victims of childhood sexual abuse (estimated final sample of 30–50 patients).

Inclusion criteria will be the following:Women over 18 years of age.Follow-up in the ACASI association or similar.Present symptoms related to the post-traumatic sequelae of having experienced sexual abuse as a child.The traumatic experience is accessible to the participant’s explicit memory.The participant has shared her experience in at least a containment context and can talk about it.

Exclusion criteria will be the following:Severe mental illness. Extreme scores in both the personality questionnaire and the psychopathology questionnaire in the indicators of global severity, paranoid ideation, and psychoticism.Any addiction problems with alcohol or other substances at the time of the assessment that may interfere with adherence to treatment and group dynamics.Being currently in treatment for the traumatic abuse experience.Presenting severe dissociative symptoms beyond those typical of a PTSD diagnosis. Extreme scores on the DES dissociation scale on pathological ideation items.

Both therapeutic approaches (EMDR, TF-CBT) will be conducted by two researchers of the study: Ps. 1. (General health psychologist, master in cognitive-behavioral therapy, consultant, clinic and EMDR facilitator, expert in trauma and attachment); and Ps. 2. (Psychologist specialized in clinical psychology, master in cognitive-behavioral therapy, expert in treatments for victims of violence and emotional disorders, therapist trained in EMDR level 1). The presence and active participation of both researchers in the two treatment groups will reduce the possible contamination effect caused by having different therapists in both groups.

### Study Schedule

This study started in April 2021. It is estimated that this study will be completed by the end of 2023.

## 4. Statistical Analysis

The evaluation instruments and the study variables collected will be coded and captured in databases. The results obtained from this intervention will be analyzed using the Statistical Package for the Social Sciences (SPSS) version 26 (IBM Corp., Armonk, NY, USA).

Independent samples *t*-tests and chi-square tests will be conducted to determine pre-intervention differences between participants with and without missing values on the variables studied. In addition, chi-square tests were conducted to compare the number of participants who dropped out in group 1 (TF-CBT + EMDR) with those who dropped out in group 2 (EMDR + TF-CBT) and showed that they did not differ significantly.

Before testing the change models, descriptive analyses, Pearson correlations, and multivariate and univariate analyses of variance (MANOVA, ANOVA) will be conducted on the pre-intervention scores of participants in groups 1 and 2 to identify possible differences in the baseline (T1). In addition, multivariate and univariate variance analyses (MANCOVA, ANCOVA) will be conducted to identify changes in post-intervention scores (short-term effect) while controlling for pre-intervention (covariate) scores (T1 and T2), and post-intervention scores at T3 and T4. In addition, the effect size (Cohen’s d) of each variable will be calculated to estimate the magnitude of the differences between the two groups. Multiple hierarchical regression analyses will be conducted to examine the added predictive value of any therapeutic condition (TF-CBT or EMDR) in explaining the efficacy of any intervention program.

### Ethics and Dissemination

Before its implementation, the protocol study was submitted to the Ethics Committee (CEISH) of the Valencian International University (VIU) for approval and endorsement.

Each participant will receive information about the study’s purpose and procedures and provide written consent to participate. The psychologist specialist researcher will obtain consent and assent. The data will be confidential and anonymous and will be used solely for the study. Numeric codes will link each participant’s identifying information. Data collected will be stored in a locker at the principal investigator’s workplace, and the electronic data will be password-protected on the university network computer. Any modifications to the protocol will be recorded on ClinicalTrials.gov (accessed on 30 March 2022).

## 5. Measurements

### 5.1. Satisfaction with Life Scale (SWLS)

It consists of a 5-item scale in which participants must indicate the extent to which they agree with each question, in a Likert format from 1 to 7, with scores ranging from a minimum of 5 to a maximum of 35, where higher scores indicate higher life satisfaction [49]. Life satisfaction as assessed by the SWLS indicates a degree of temporal stability (e.g., 0.54 over 4 years), although the SWLS has shown sufficient responsiveness to be potentially valuable for detecting changes in life satisfaction over the course of clinical intervention. In addition, the scale shows discriminant validity for measures of emotional well-being. The SWLS is recommended to be used alongside scales that focus on psychopathology or emotional well-being because it assesses the conscious evaluative judgement of individuals about their lives using their own criteria [50]. The scale has been validated in a sample of Spanish adolescents. The reliability analysis of the scale indicated that the Spanish version has a good internal consistency (α = 0.84).

### 5.2. Rosenberg Self-Esteem Scale (RSE)

The Rosenberg Self-Esteem Scale (RSES) [51,52] has been a popular self-esteem indicator. Self-esteem is one of the psychological variables that is severely affected in adult victims of childhood sexual abuse [53].The RSES is used worldwide in a wide range of studies and research [54,55,56]. The RSES assesses the feeling of satisfaction that a person has about himself/herself. It consists of ten items focusing on general feelings of self-respect and self-acceptance. The total score ranges from 10 to 40 points, with a distinction between low (scores ≤ 25), medium (26–29), and high (≥30) self-esteem. The test–retest reliability is r = 0.85, and the alpha coefficient for internal consistency is very high [57], α = 0.92. The scale has good reliability and validity in the Spanish clinical population [58].

### 5.3. Symptom Checklist-90-Revised (SCL-R) 

The SCL-90-R [59] is most commonly used by mental health professionals to assess psychological symptoms and monitor patient progress during and after treatment [60]. Symptomatology in several areas of function is common in people who were sexually abused in childhood. Therefore, an instrument such as the SCL-90-R is useful to classify symptoms by clusters, especially if there is no clear pattern of post-traumatic stress, but there is significant symptomatology resulting from the trauma [61].

This self-administered questionnaire presents 90 items describing symptoms and requires the individual to indicate on a Likert scale between 0 (not at all) and 4 (very or extremely) to what extent he/she feels bothered by each of the symptoms described. In the Spanish adaptation [62], reliability showed a Cronbach’s alpha for the entire scale (0.97).

### 5.4. Post-traumatic Stress Disorder Symptom Severity Scale according to the DSM-5 (EGS-R)

EGS-R [63] is a 21-item scale, based on the DSM-5 diagnostic criteria, to assess the presence and severity of symptoms. It is a structured evaluation instrument, in Likert format (from 0 to 3 according to the frequency and intensity of symptoms). The EGS-R has shown adequate psychometric properties in the Spanish population with a high internal consistency (α = 0.91).

Women who report a history of childhood sexual abuse frequently experience PTSD symptoms correlated with depression severity, anxiety, self-esteem, and other clinical symptomatology [64], making it useful to have a scale based on the DSM-5 diagnostic categories to assess the presence of PTSD [65].

### 5.5. DSM-5 Personality Inventory—Brief Version (PID-5-BF) Adults

This is an abbreviated version of the 220-item PID5 Inventory [66] consisting of 25 items assessing the five dominant dysfunctional personality traits proposed in section III of the fifth version of the DSM-5. There are five dimensions with five items each: negative affectivity, detachment, antagonism, disinhibition, and psychoticism. The items are evaluated according to the degree of agreement with each statement on a 4-point scale (0 being “fairly untrue” and 3 being “fairly true”). 

Sexual abuse appears to have adverse long-term consequences for many victims. This may have implications for their ability to relate with others and for the friendships they will build as adults [67]. To address the frequent re-victimization in this population, it was decided to assess personality factors [68].

The Spanish version of the extended scale showed a mean internal consistency of α = 0.86 in the clinical sample and α = 0.79 in the control sample [69].

### 5.6. Scale of Emotional Regulation Difficulties (DERS) 

The DERS [70] assesses different aspects of the emotional regulation process: emotional dyscontrol, daily interference, emotional inattention, emotional confusion, and emotional rejection. Emotion regulation difficulties are common in people with PTSD [71]. The total number of items used in the final adapted version was 28. The Spanish version of the DERS scale offers good psychometric properties with a Cronbach’s α = 0.93 for the total scale [72].

### 5.7. Dissociative Symptom Scale (DSS) 

The Dissociative Symptom Scale (DSS) [73] was developed to assess moderately severe levels of depersonalization, derealization, gaps in awareness or memory, and dissociative re-experiencing that would be relevant to a wide range of clinical populations, especially those who have been victims of childhood sexual abuse and post-traumatic symptomatology [74].

DSS is a self-administered 28-item scale designed to measure dissociative symptomatology. Items are scored according to the frequency of each dissociative experience on a range from 0 to 100, where 0 represents “never” and 100 represents “always”. The DES has good psychometric properties, with a Cronbach’s α of 0.91 in the Spanish validation [75].

### 5.8. Scale of Satisfaction with the TREATMENT received. (CRES-4): Spanish Version 

Being able to measure patient satisfaction at the end of treatment has become essential nowadays, not only from the patient’s perspective, who will feel heard, but also for the therapist and the center where the treatment is carried out. In this sense, the CRES-4 [76] is an easy-to-use instrument that responds to this need in the Spanish-speaking psychotherapy field.

It is a 4-item scale designed to assess the patient’s degree of satisfaction with the therapy received, the extent to which they feel their main problem has been resolved and the perceived change in their emotional state from pretreatment to post-treatment. The overall score is intended to reflect the effectiveness of the treatment according to the patient [77].

### 5.9. Ad Hoc Registry for General Sociodemographic and Clinical Variables

Participants will be asked for information about age, gender, marital status, educational level, employment status and work history, family socioeconomic status, previous psychological/psychiatric treatments, somatic and psychiatric illnesses, and descriptive elements of childhood sexual abuse.

## 6. Protocols of Treatment

This section explains both protocols, which are summarized in Table 2.

## 7. Trauma-Focused CBT-Based Treatment

An online innovative group intervention protocol focused on trauma has been designed, adapting the protocol of Cohen [78] for complex trauma.

The TF-CBT is an evidence-based therapeutic approach to improve symptoms of PTSD as well as affective or cognitive and behavioral problems. The treatment will be composed of three phases of 2–3 sessions each.

### 7.1. Phase 1: TF-CBT Coping Skills for Complex Traumas

The initial phase has three objectives: (1) Establishing a trusting relationship with the psychologist and developing safer and more effective self-regulation skills; (2) reinforcing safety, the psychologist encourages the development of safety by helping the woman to identify the presence of people in her current life with whom she now relates as safe for herself. (3) Psychoeducation provides information about the impact of the trauma, reminders of the trauma, and hope for recovery [79]. (4) Relaxation skills, affective and cognitive coping, mindfulness.

### 7.2. Phase 2: Trauma Narration and Complicated Trauma Processing

The development of the trauma narrative is an interactive process between the woman and the psychologist; it is an actual process in sessions. In most cases, the final narrative “product” reflects only a small part of this process. Trauma processing involves women describing complex traumatic experiences from the past and coming to new insights about the meaning of these life experiences. Women must identify and carefully examine the impact of core beliefs related to the underlying issue.

Even after adopting a new, more functional belief, if it is not yet sufficiently integrated into the cognitive map, a return to previous maladaptive beliefs may occur. In some cases, these women, when faced with new, highly stressful experiences, may suffer a reactivation and a return to the cognitive functioning driven by the previous dysfunctional beliefs. In these cases, the psychologist prepares them for the possible reactivation and will encourage them to try the new, more functional belief repeatedly. Live exposure to memories of the trauma.

Development of a hierarchy of feared stimuli and a gradual exposure schedule. Exposure is gradual and does not progress from one situation to another until it is well-assimilated and processed. This provides opportunities to gain skills and confidence.

### 7.3. Phase 3: Treatment Consolidation and Closure

After processing the trauma, the final phase of TF-CBT encourages the gradual transfer of communication between the psychologist and the woman, establishing further trusting and positive relationships with other people in her environment, and maintaining safety in everyday life. Each woman can share with the others the progress she feels she has achieved on an individual level. This can be reviewed in a follow-up session after one month to test and repeatedly apply what they have previously learned in the TF-CBT treatment as they try to establish safety and develop appropriate relationships in real-life situations.

## 8. Trauma-Focused EMDR-Based Treatment

A trauma-focused group intervention protocol was designed, after studying group EMDR protocols.

Jarero and Artigas developed an EMDR group therapy protocol (EMDR-IGTP) in response to the high number of requests for mental health care natural catastrophes [80]. EMDR-IGTP has been used with children and adults in different places of the world and many studies report its effectiveness with children and adults in response to disasters, ongoing war trauma, ongoing geopolitical crisis, displacement of war refugees, work accidents, and severe IPV [81].

The Group Traumatic Event Protocol (G-TEP) was developed by Elan Shapiro to build a group protocol that incorporates the strengths of EMDR and the AIP model and closely resembles the individual protocol. G-TEP is an adaptation of the Recent Traumatic Events Protocol (R-TEP) to be used with groups of adults, older children, and adolescents who have had recent traumatic experiences or long-term life-changing events with ongoing consequences that are not necessarily recent. It requires a simplified, structured, worksheet format suitable for rapid assimilation and effective use with groups or individuals [82].

The result is an innovative eight-phase EMDR protocol that will be applied in a telematic group format. Phases 3 to 7 will be conducted during sessions 3 to 8. If a participant becomes destabilized during the group work, they will be treated in a private virtual room with the Stabilization Protocol for Acute Stress (EMDR-PESEA) [83].

Phase (1). Client history before session 1.

Phase (2). Preparation for the treatment of the traumatic event, with psychoeducation and regulation strategies. The resources to be used in the subsequent phases of reprocessing are also prepared. Introducing the four-elements exercise and drawing a resource or symbol of a reassuring place, and installing positive feelings with EBL.

Phases (3)–(6).

Phases 3 to 6 are carried out following the G-TEP worksheet. It provides a sense of security (safe place, past resource, desired future-PC, timeline), control structure, order, and differentiation of past and present (moving concretely between the past danger and the present safety). The EMD strategy provides containment boundaries to present T-Episode.

Phase (7). Closure of the session.

A group debriefing of the experience will take place and some of the stabilization exercises prepared in phase 2 (four elements and the container) will be carried out. This will allow the patient to exit the associative channels that have been activated by focusing on the traumatic event and move on to the closure phase.

Phase (8). Re-evaluation.

This phase will take place immediately after the group intervention. It assesses which participants may need individual attention and which may need further evaluation to identify the nature and extent of their symptoms.

## 9. Conclusions

TF-CBT and EMDR are therapeutic approaches that have been empirically tested, and their therapeutic efficacy has been demonstrated; however, studies such as the one proposed are still needed to compare both therapies. Our study is noteworthy because it offers two novel treatment programs that bring substantial and novel improvements to existing approaches in TF-CBT and EMDR for working with female victims of childhood sexual abuse. In addition, the effect and efficacy of the application of both treatments, separately and their cumulative effect, will be analyzed comparatively.

Another significant contribution of our protocol is that the interventions will be carried out entirely in an online group format (MHealth). This will make it possible to offer the alternative of online treatment to victims of childhood sexual abuse within this risky context of social distancing due to the pandemic. It is crucial to offer therapeutic resources at this time, as victims of sexual abuse are potentially at risk of being re-victimized and may be more vulnerable to experience a worsening of symptoms after the outbreak of COVID-19, compared to people without psychiatric problems [84].

Online and group-based treatment has the advantage of greater accessibility and can reach more participants, even in remote areas or people isolated in their homes where access to health services is limited. In this way, even when the COVID-19 pandemic has been brought under control, the research evidence and digital infrastructure will be in place to offer online treatment to more participants, even in low-income or developing countries where access to psychological treatment is difficult, as online interventions have shown positive results in reducing anxiety and depression, among other symptoms [85].

This study will also assess adherence to online treatment which, given the current context of the COVID-19 pandemic, will be fundamental in contributing to research on telematic psychological intervention and assessing its strengths and weaknesses.

We expect to provide evidence on the benefits of both psychotherapies in reducing the symptomatology presented by this population and which hinders their functioning in different areas of adult life.

As noted in the introduction, results in the literature indicate that both models of psychotherapy significantly reduce PTSD and dissociative symptoms, behavioral problems, anger, depression, and anxiety in these women as well as improve emotion regulation strategies and self-esteem. Furthermore, after undergoing one of these treatments, their well-being will incorporate a healthier perspective linked to higher self-esteem, a better perception of their reality, and greater satisfaction with life. When presenting a study that integrates both the comparative analysis of the separate benefits of each type of therapeutic model (EMDR, CBT-TF) and the analysis of the effect of the amplification of both models combined sequentially, it is intended to contribute to the development of therapeutic protocols that incorporate those elements that are clearly more favorable to the well-being of these women.

## Figures and Tables

**Figure 1 ijerph-19-07468-f001:**
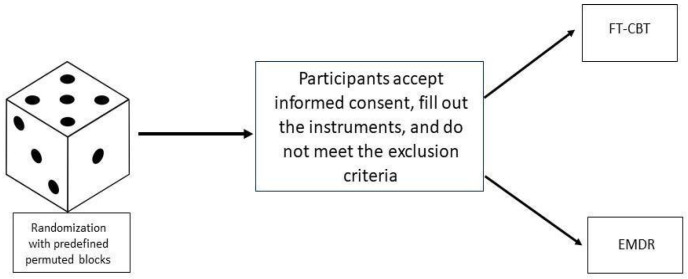
Randomization process.

**Figure 2 ijerph-19-07468-f002:**
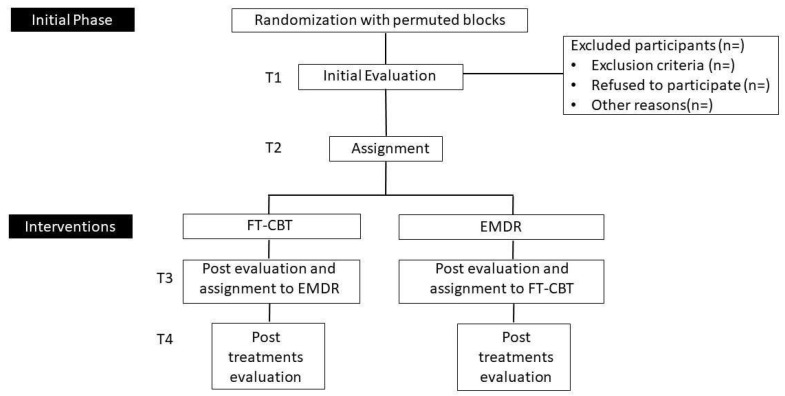
Study design FT-CBT and EMDR.

**Table 1 ijerph-19-07468-t001:** Example template of recommended content for the schedule of enrolment, interventions, and assessments.

	STUDY PERIOD
	Enrolment	Allocation	Post-Allocation	Close-Out
**TIMEPOINT ***	** *−t1* **	**0**	** *t1* **	** *t2* **	** *t3* **	** *t4* **	
**ENROLMENT:**	X					
**Eligibility** **screen**		X				
**Informed** **consent**			X			
**Filling the** **questionnaires**						
**Allocation**		X				
**INTERVENTIONS:**						
EMDR + TF-CBT		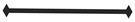		X	X
TF-CBT + EMDR		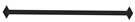		X	X
**ASSESSMENTS:**						
** *Satisfaction with life scale (SWLS)* **			X	X	X	X
** *Rosenberg Self-Esteem Scale* ** ** *(RSE)* **			X	X	X	X
** *Symptom Checklist-90-Revised (SCL-R)* **			X	X	X	X
** *Post-traumatic Stress Disorder Symptom Severity Scale according to* ** ** *the* ** ** *DSM-5* ** ** *(EGS-R)* **			X	X	X	X
** *DSM-5 Personality Inventory—Brief Version (PID-5-BF)* ** ** *Adults* **			X			X
** *Scale of emotional regulation difficulties (DERS)* **			X	X	X	X
** *Dissociative Symptom Scale (DES)* **			X	X	X	X
** *Scale of satisfaction with the treatment received. (CRES-4)* **					X	X	
** *Ad-hoc registry for general sociodemographic and clinical* ** ** *variables* **			X				

Recommended content can be displayed using various schematic formats. See SPIRIT 2013 Explanation and Elaboration for examples from protocols. * List specific timepoints in this row. Intervention A: EMDR + TF-CBT. The group receives the EMDR treatment first, and afterwards, the TF-CBT Intervention B: TF-CBT + EMDR. The group receives the TF-CBT treatment first, and afterwards, the EMDR.

**Table 2 ijerph-19-07468-t002:** Treatment protocols.

Treatment Protocols	
**Trauma-Focused CBT-based treatment** Adaptation of the protocol of Cohen for complex trauma. TF-CBT is an evidence-based therapeutic approach for treating traumatized patients. Generate an improvement of symptoms of PTSD, dissociative, affective, or cognitive and behavioral problems. Eight weekly one-hour online group sessions per week Three phases of 2–3 sessions each. Phase 1: TF-CBT Coping Skills for Complex Traumas. Objectives: Establishing a trusting relationship and self-regulation skills; reinforcing safety; psychoeducation; relaxation skills, mindfulness affective and cognitive coping. Phase 2: Trauma Narration and Complicated Trauma Processing. Objectives: The development of the trauma narrative; identify and examine the impact of core beliefs; development of a hierarchy of feared stimuli and a gradual exposure schedule. Live exposure to trauma memories. Phase 3: Consolidation and completion of treatment. Objectives: After processing the trauma, share the individual progress achieved with others; follow-up sessions; ensure safety and develop appropriate relationships in real-life situations.	**Trauma-Focused EMDR treatment** Adaptation of Jarero and Artigas’s EMDR group therapy protocol (EMDR-IGTP) and Elan Shapiro’s Traumatic Event Protocol (G-TEP) for complex trauma. EMDR is an evidence-based therapeutic approach for treating traumatized patients. Generate an improvement in PTSD symptoms, dissociative, affective, or cognitive and behavioral problems. Eight weekly one-hour online group sessions per week Eight-phase EMDR protocol. Phases 3 to 7 will be conducted during sessions 3 to 8. Phase 1: Objective: Client history before session 1. Phase 2: Objectives: Preparation for the treatment of the traumatic event, with psychoeducation and regulation strategies, calming place, and setting up positive feelings with EBL. Phases 3–6: Objectives: performed following the G-TEP worksheet. Sense of security (safe place, past resource, desired future-PC, timeline) control structure, order, differentiation of past and present. Phase 7: Closing of the session. Objectives: A group debriefing of the experience and conducting some of the stabilization exercises. Phase 8: Objectives: Re-evaluation. After the group intervention, assess any need for individual attention.

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
