# Peer review of "Psychological Intervention in Women Victims of Childhood Sexual Abuse: An Open Study—Protocol of a Randomized Controlled Clinical Trial Comparing EMDR Psychotherapy and Trauma-Based Cognitive Therapy"

_ijerph, 2022, doi:10.3390/ijerph19127468_

Round 1

Reviewer 1 Report

The idea of presenting a protocol that might be used for research is interesting. The challenge is that we have no idea that the protocol is valid until the work has been carried out. At the moment, the paper reads like an extended ethics application. 

There were areas that I wondered about:

  1. Why were the measurements chosen? How do they add to answering the research question?
  2. How is recruitment being done?
  3. In the CBT and the EMDR groups, will the same therapist work with all clients in either the CBT or EMDR? I am wondering about how therapist bias is being managed? 
  4. In terms of the EMDR approach outline for use in the research been validated? I ask this as the CBT approach has a significant body of work while the EMDR approach seems novel. How might that impact the research design?
  5. Are there incentives for participation?
  6. In the abstract, starting on line 44 with the word ethics - does not need to be in the abstract.
  7. Line 319 - women tend to revert - what is the basis for that statement?
  8. How were the agencies you will be working with chosen?

Overall, I find publishing this sort of paper has merit post-hoc when there is evidence that the protocol has been effective.

Author Response

Dear Reviewer we would like to thank for their effort and time invested in reviewing our manuscript.  We have tried to include the suggestions discussed and presented below, and we hope that after they have been made, we can continue with the publication process of the journal. We appreciate the time and effort spent on the revision because we feel that the manuscript has been greatly improved. Below is the response to each of the suggestions made, the changes made have been marked in the manuscript in red.

Reviewer: 1
Comments to the Authors

  • English language and style are fine/minor spell check required  
    Response: Thank you for the suggestion, we have reviewed the English language.
  • Why were the measurements chosen? How do they add to answering the research question?

Response: In order to create a more adequate explanation we have incorporated in the introduction section information regarding the importance of the variables that have been evaluated in this research, including studies from the scientific literature that point out the importance of these variables as fundamental elements of the psychological clinic in the study population, as well as the object of attention to evaluate change after psychological treatment in these women.

  • How is recruitment being done?

Response: To improve this section, we have expanded the information on recruitment.

  • In the CBT and the EMDR groups, will the same therapist work with all clients in either the CBT or EMDR? I am wondering about how therapist bias is being managed? 

Response: In response to your suggestion, we have included the following paragraph in the text to clarify this point:"Both therapeutic approaches will be carried out by two of the researchers of the study: Ps.1. (General Health Psychologist, Master in cognitive-behavioural therapy, Consultant, Clinician and EMDR facilitator, expert in trauma and attachment); and Ps.2. (Psychologist specializing in Clinical Psychology, Master in cognitive-behavioural therapy, expert in treatments for victims of violence and emotional disorders. Therapist with EMDR level 1 training). The presence and active participation of the two researchers in the two treatment groups will buffer the possible contamination effect generated by the presence of different therapists in both groups".

  • In terms of the EMDR approach outline for use in the research been validated? I ask this as the CBT approach has a significant body of work while the EMDR approach seems novel. How might that impact the research design?

Response: Following your indications, we have incorporated in the introduction section, data from the scientific literature that provide evidence of the scientifically proven importance of both treatments (EMDR and CBT-TF) as treatments of choice for the treatment of people with clinical trauma, in particular, for childhood sexual abuse trauma. We also incorporate information regarding the history of both models of psychotherapy.

  • Are there incentives for participation?

Response: For the avoidance of doubt, we have included in the text the following statement: “all participated voluntarily without any incentive.”

  • In the abstract, starting on line 44 with the word ethics - does not need to be in the abstract.

Response: Thank you, we have removed this part of the abstract.

  • Line 319 - women tend to revert - what is the basis for that statement?

Response: To clarify this point, we have re-write this part of the text.

How were the agencies you will be working with chosen?

Response: To clarify this, we have included extra information about this.  

Reviewer 2 Report

  1. In the section of introduction, I suggest the author could provide more information about the reasons why to employ TF-CBT and EMDR to solve the researching problems. In addition, I suggest the author could provide briefly history or development about integrate TF-CBT and EMDR into interventions.
  2. In the section of method, I suggest the author could provide more information about CONSORT statement and explain the advantages and useful effect about RCT in this study.
  3. In the section of measurement, I suggest the author could provide more theoretical statements to explain why to employ these scales and should provide the results of validity about these scales.
  4. In the end, I suggest the author could reorganize the results and discussion, such as providing some tables or figures to help readers to understand the researching results.
  5. In the section of conclusion, I cannot find the theoretical or practical values in this manuscript. I suggest the author could focus on the different effects of interventions to discuss the appropriate or useful strategies to help these people.

Author Response

Dear Reviewer we would like to thank for their effort and time invested in reviewing our manuscript.  We have tried to include the suggestions discussed and presented below, and we hope that after they have been made, we can continue with the publication process of the journal. We appreciate the time and effort spent on the revision because we feel that the manuscript has been greatly improved. Below is the response to each of the suggestions made, the changes made have been marked in the manuscript with track changes.

  • English language and style are fine/minor spell check required  
    Response: Thank you for the suggestion, we have reviewed the English language.
  • In the section of introduction, I suggest the author could provide more information about the reasons why to employ TF-CBT and EMDR to solve the researching problems. In addition, I suggest the author could provide briefly history or development about integrate TF-CBT and EMDR into interventions.

Response: Following your indications, we have incorporated in the introduction section, data from the scientific literature that provide evidence of the scientifically proven importance of both treatments (EMDR and CBT-TF) as treatments of choice for the treatment of people with clinical trauma, in particular, for childhood sexual abuse trauma. We also incorporate information regarding the history of both models of psychotherapy.

  • In the section of method, I suggest the author could provide more information about CONSORT statement and explain the advantages and useful effect about RCT in this study.

Response: To clarify this point, we have included some more information about this.  

In the section of measurement, I suggest the author could provide more theoretical statements to explain why to employ these scales and should provide the results of validity about these scales.

Response: Following your indications, we have incorporated in the measurement section, data pointing out the importance of the chosen assessment instruments as well as their validation data

  • .In the end, I suggest the author could reorganize the results and discussion, such as providing some tables or figures to help readers to understand the researching results.

Response: Following your suggestion, we have created and incorporate a new figure, that it is a table that helps to better explain the objectives of the different treatment protocols and the results we are intended to achieve.

  • In the section of conclusion, I cannot find the theoretical or practical values in this manuscript. I suggest the author could focus on the different effects of interventions to discuss the appropriate or useful strategies to help these people.

Response: Thank you for this suggestion, we have expanded on this kind of information in the conclusion section.

Round 2

Reviewer 2 Report

The author mostly incorporated my suggestions and revised the manuscript. The manuscript could be published in this form for the journal.